# RCSegNeXt: Efficient multi-scale ConvNeXt for rectal cancer segmentation from sagittal MRI scans

| | |
|---|---|
| **Wang Bo**[1,2] | wang_bo@mail.ustc.edu.cn |
| **Ting Xue**[3] | 849638196@qq.com |
| **Leyang Pan**[3] | 15058830156@163.com |
| **Dingfu Huang**[1,2] | hdf0824@mail.ustc.edu.cn |
| **Yi Xiao**[3] | czyyxiaoyi@163.com |
| **Li Fan**[3] | fanli0930@163.com |
| **Zaiyi Liu**[4,5] | Liuzaiyi@gdph.org.cn |
| **Shiyuan Liu**[3] | radiology_cz@163.com |
| **S Kevin Zhou**[*1,2,6] | skevinzhou@ustc.edu.cn |

[1] *School of Biomedical Engineering, Division of Life Sciences and Medicine, University of Science and Technology of China, Hefei, Anhui, 230026, P.R.China*

[2] *Suzhou Institute for Advanced Research, University of Science and Technology of China, Suzhou, Jiangsu, 215123, P.R.China*

[3] *Department of Radiology, Second Affiliated Hospital of Navy Medical University, Shanghai 200003, China.*

[4] *Department of Radiology, Guangdong Provincial People's Hospital, Guangdong Academy of Medical Sciences, Guangzhou, China*

[5] *Guangdong Provincial Key Laboratory of Artificial Intelligence in Medical Image Analysis and Application, Guangdong Provincial People's Hospital, Guangdong Academy of Medical Sciences, Guangzhou, China*

[6] *State Key Laboratory of Precision and Intelligent Chemistry, University of Science and Technology of China, Hefei, Anhui 230026, China*

**Editors:** Accepted for publication at MIDL 2025

## Abstract

Rectal cancer remains a critical global health challenge, significantly contributing to morbidity and mortality worldwide. Magnetic resonance imaging (MRI) in a sagittal plane offers distinct advantages for rectal cancer diagnosis by providing detailed visualization of the rectum and its surrounding anatomy. However, automated segmentation of the rectum and associated tumors remains difficult due to tumor heterogeneity and complex anatomical structure, which necessitate multi-scale feature extraction. This study proposes RCSegNeXt, a novel non-uniform pure-convolutional rectal cancer segmentation architecture that combines shallow anisotropic stages with deep isotropic stages. The anisotropic stages leverage AniNeXt blocks, designed with customized convolutional kernels and pooling operations to address the uneven spatial resolution inherent in MRI data. In the isotropic stages, an IsoNeXt block with a Scale-Aware Integration Module (SAIM) enables efficient multi-scale feature fusion by directing information flow through constrained pathways. This design enhances computational efficiency while achieving superior segmentation accuracy. Experiments on two in-house rectal cancer datasets and a publicly-available prostate dataset demonstrate the proposed method's state-of-the-art performances. Code is available at GitHub.

**Keywords:** Rectal cancer, image segmentation, anisotropic MRI, multi-scale.

## 1. Introduction

Colorectal cancer, a highly aggressive digestive system malignancy, ranks as the third most commonly diagnosed cancer and the second leading cause of cancer-related deaths worldwide (Sung et al., 2021). Notably, a significant proportion of these cases are localized to the rectal region, with this proportion reaching nearly 50% in China (Qu et al., 2022). Among the three orthogonal planes used in magnetic resonance imaging (MRI) for rectal cancer, the sagittal plane offers distinct diagnostic advantages, enabling comprehensive visualization of the rectum, tumor, and anus in a single image (Shen et al., 2023). It also provides a detailed depiction of the rectum's relationship with the peritoneum and adjacent organs, making it indispensable for accurate tumor T staging. Despite its critical role, precise segmentation of the rectum and tumor in sagittal MRI remains underexplored. This study aims to fill this gap by *developing 3D segmentation methods for the rectum and tumor in sagittal MRI*.

The automatic segmentation of the rectum and tumor poses unique challenges, primarily due to the need for incorporating multi-scale information. These challenges arise from two main factors: (1) the diverse tumor sizes caused by tumor heterogeneity and individual variations, as demonstrated in the first two cases of Fig. 1, and (2) the necessity of multi-scale information to accurately identify tumors, as demonstrated in the last two cases of Fig. 1. In the MRI XY plane, tumor diagnosis relies not only on the immediate vicinity of the tumor but also on the surrounding intestinal wall. In the Z plane, precise tumor delineation necessitates the analysis of multiple adjacent slices. Therefore, integrating multi-scale features is essential for accurate segmentation. While various approaches aim to improve multi-scale representation (Ronneberger et al., 2015; Lin et al., 2017; Bo et al., 2022), their layer-wise operations remain relatively coarse, treating features at different resolutions as separate scales. Res2Net (Gao et al., 2019) enhances multi-scale representation by partitioning convolutions into smaller groups, enabling finer-grained processing of features with diverse receptive fields. However, Res2Net has two primary limitations: firstly, it is less efficient compared to transformer-inspired architectures such as ConvNeXt (Liu et al., 2022), and secondly, although multi-scale features are captured, they are processed in an unstructured flow across scales, resulting in suboptimal multi-scale fusion.

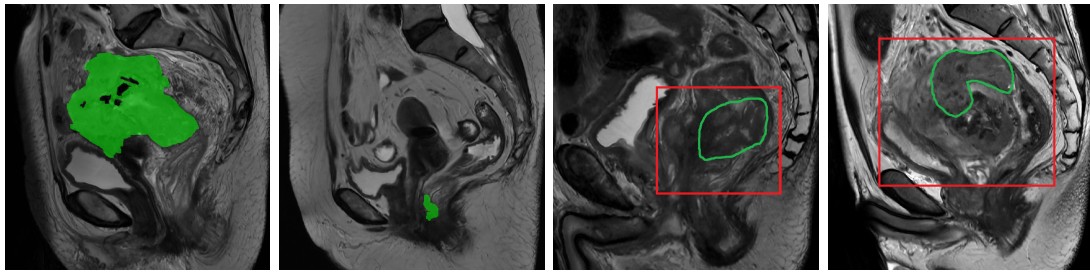

Figure 1: Slices from different patients. Green regions indicate tumors, green curves mark areas adjacent to tumors, red boxes highlight regions critical for tumor diagnosis.

In this paper, we propose a novel approach named RCSegNeXt for segmenting the rectum and tumor in sagittal MRI by employing a non-uniform architecture composed of two components. The first component utilizes anisotropic stages with AniNeXt blocks, which

incorporate anisotropic convolutions and max-pooling operations to mitigate the challenge of excessive slice thickness in MRI. The second component enhances multi-scale feature representation through the proposed IsoNeXt block, which introduces two key innovations: (1) a ConvNeXt-like transformer structure for enhanced efficiency, and (2) a Scale-Aware Integration Module (SAIM) to regulate the direction of information flow. The SAIM effectively integrates multi-scale features both intra- and inter-scale, providing a more effective feature fusion than unconstrained methods. Our main contributions are as follows:

- To the best of our knowledge, we are the first to achieve automatic end-to-end 3D segmentation of the rectum and tumor in sagittal plane MRI scans for rectal cancer.

- We introduce a non-uniform architecture that addresses the challenge of thick MRI slices, incorporating the IsoNeXt block in isotropic stages to leverage multi-scale features effectively. The SAIM within the IsoNeXt block specifically imposes constraints on the direction of information flow.

- Extensive experiments on two in-house rectal cancer datasets and a publicly-available prostate dataset demonstrate the effectiveness of our approach.

## 2. Related Work

Medical image segmentation has made substantial progress, largely driven by innovations in deep learning (Liu et al., 2022). Initially, convolutional neural networks (CNNs) dominated, with UNet's symmetric encoder-decoder architecture setting the standard. Subsequent developments extended UNet with variants like VNet (Milletari et al., 2016), UNet++ (Zhou et al., 2018). Notably, nnUNet (Isensee et al., 2021) marked a significant leap through self-configuring workflow optimization. The advent of vision transformers (ViTs) (Dosovitskiy et al., 2020) catalyzed a paradigm shift, yielding hybrid architectures like TransUNet (Chen et al., 2021), CoTr (Xie et al., 2021) that explicitly integrate CNN-Transformer interactions, along with transformer-dominant frameworks like UNETR (Hatamizadeh et al., 2022), SwinUNETR (Hatamizadeh et al., 2021), which retain CNN-based decoders for feature reconstruction. Transformer-only approaches like SwinUNet (Cao et al., 2022) further showcased the power of transformers. More recently, state-space models (SSMs) have emerged as computationally efficient alternatives, with SegMamba (Xing et al., 2024) replacing self-attention with Mamba blocks for enhanced efficiency. Concurrently, foundation models like SAM (Kirillov et al., 2023), SAM2 (Ravi et al., 2024) and MedSAM (Ma et al., 2024) have shown impressive generalization capabilities in segmentation via prompt engineering. Despite these advances, rectal cancer segmentation in MRI still demands further refinement beyond current architectures.

## 3. Method

### 3.1. Overview

The architecture of the proposed method is depicted in Fig. 2. As illustrated, the design incorporates a non-uniform structure consisting of two main components: the shallow anisotropic stages and the deep isotropic stages.

**Anisotropic stages.** In the anisotropic stages, Ani(sotropic)NeXt blocks with anisotropic convolutions of kernel size 1×3×3 are employed. Max-pooling operations are also anisotropic, with a kernel size of 1×2×2. This design is specifically tailored to address the characteristics of rectal cancer MRI scans, which often feature substantially thick slices. For instance, the median spacing in the in-house Dataset A is [4.000, 0.875, 0.875], while Dataset B exhibits [3.300, 0.234, 0.234]. Applying conventional convolutions uniformly across all three axes would disproportionately emphasize features on the high-resolution plane while inadequately capturing axial-direction features due to the uneven spatial resolution. Anisotropic kernels are repeated until the spacing among the three axes becomes approximately uniform, with the stopping criterion defined as $spacing_z/spacing_{x(y)} < 2$.

**Isotropic stages.** In the isotropic stages, features are processed uniformly across all dimensions. To enhance feature extraction, multi-scale Iso(tropic)NeXt blocks are employed at this stage. The IsoNeXt block utilizes 3×3×3 convolutions, and max-pooling operations have a kernel size of 2×2×2. It is important to note that the IsoNeXt blocks are exclusively used during the isotropic stages and not in the shallow anisotropic stages. This decision is based on how multi-scale features are constructed in an IsoNeXt block by stacking small groups of convolutions to achieve varying receptive fields. In the anisotropic stages, the convolution kernel size along the axial direction is fixed at 1. Consequently, stacking several 1× convolutions does not expand the receptive field, rendering IsoNeXt blocks ineffective in these stages. Detailed specifications of the AniNeXt and IsoNeXt blocks are provided in the subsequent sections.

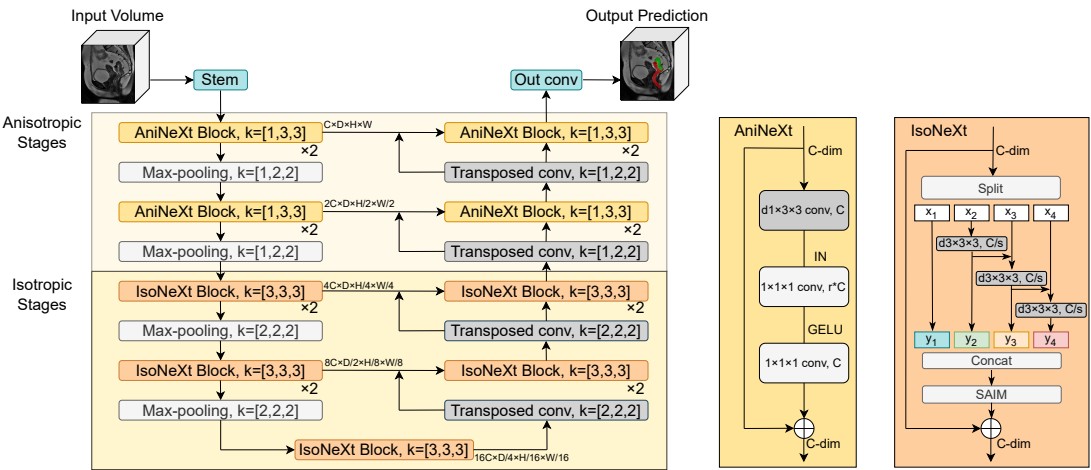

Figure 2: Overview of the proposed architecture, where k denotes kernel size, d-conv denotes depth-wise convolution, r is the expand ratio, s is the number of scales.

### 3.2. AniNeXt Block

Inspired by the design principles of vision transformers (Dosovitskiy et al., 2020; Liu et al., 2021), ConvNeXt integrates transformer-inspired elements into convolutional networks,

achieving superior performance compared to traditional ResNet (He et al., 2016) architectures. Extending the ConvNeXt framework, we propose the AniNeXt block for anisotropic stages, as illustrated in Fig. 3(a). The AniNeXt block comprises two main components: (1) Depthwise Convolution Layer: This layer employs depthwise convolution with a kernel size of $1{\times}3{\times}3$. We opt for a relatively conservative kernel size of 3 instead of larger kernels to avoid exacerbating information imbalances across different dimensions. Instance Normalization (IN) replaces Layer Normalization (LN) from ConvNeXt to accommodate the high variability in medical images, particularly in tumor characteristics. (2) Inverted Bottleneck: The inverted bottleneck structure includes two $1{\times}1{\times}1$ convolutional layers with a GELU activation function between them. The first layer expands the channel dimension from $C$ to $r \times C$, where $r$ is the expansion rate, while the second compresses it back to $C$.

### 3.3. IsoNeXt Block

In this section, we introduce the design of the IsoNeXt block, which extends the Res2Net block by modernizing it with a transformer-style architecture and incorporating a Scale-Aware Integration Module (SAIM) to enhance its ability to learn multi-scale features.

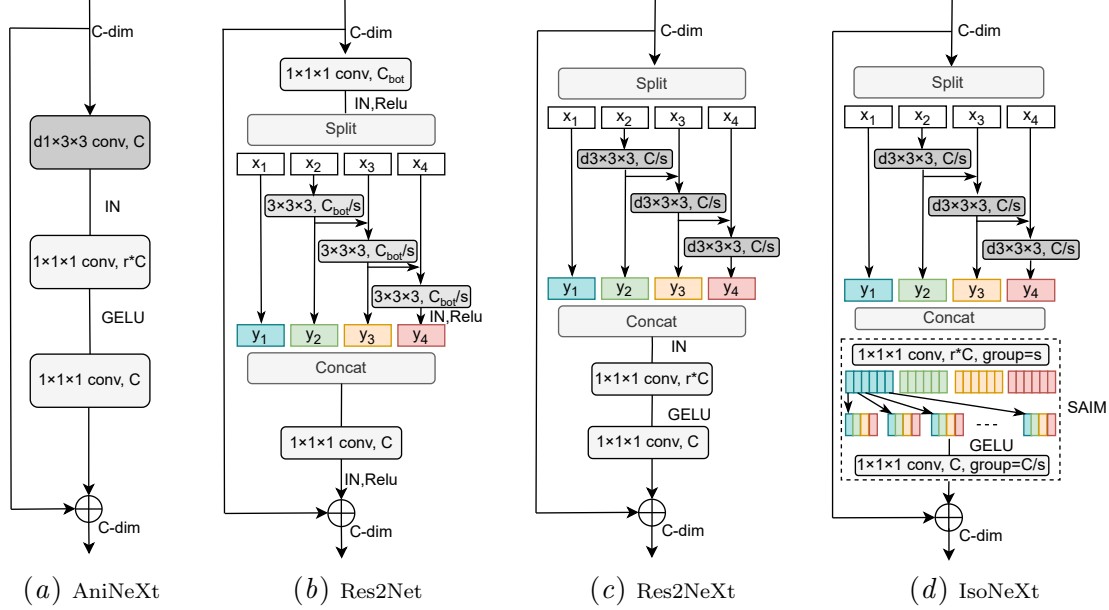

Figure 3: Comparison of different architectures, bot is short for bottleneck.

**Revisiting the Res2Net block.** The structure of the Res2Net block, illustrated in Fig. 3(b), adopts a ResNet-like bottleneck design. Following the initial $1{\times}1{\times}1$ convolution, the input feature $\mathbf{x}$ is divided into $s$ groups $\{\mathbf{x}_1, \mathbf{x}_2, \ldots, \mathbf{x}_s\}$, where $s$ represents the scale. All groups except $\mathbf{x}_1$ are processed using smaller convolutional operations, progressively capturing more fine-grained patterns. Importantly, a residual connection links each $\mathbf{y}_i$ (the convolutional output of $\mathbf{x}_i$) to $\mathbf{x}_{i+1}$. This architecture allows higher-index groups to capture larger receptive fields, thus facilitating the learning of multi-scale features, as indicated by the varying colors in Fig. 3(b), which represent features at different scales.

**Modernizing the Res2Net block.** To incorporate modern design principles from ConvNeXt with the multi-scale capabilities of Res2Net, we propose a redesigned Res2Net block, termed Res2NeXt, as illustrated in Fig. 3(c). The key modifications include: (1) replacing all $3\times3\times3$ convolutions with depthwise $3\times3\times3$ convolutions; (2) transforming the bottleneck pipeline from a $1\times1\times1$ convolution-group convolution-$1\times1\times1$ convolution structure to a depthwise group convolution-inverted bottleneck architecture; and (3) removing redundant normalization and activation layers, retaining only an IN layer after the depthwise group convolution and a GELU activation following the expansion $1\times1\times1$ layer in the inverted bottleneck. These updates result in a more efficient and streamlined implementation, while maintaining the multi-scale feature extraction capabilities.

**Scale-Aware Integration Module.** While Res2Net and Res2NeXt blocks can effectively capture multi-scale features, they do not fully exploit them. After split-group convolution-concatenation operations, the features are sequentially organized by scale, as depicted in Fig. 3(d), with distinct colors representing different scales. However, applying a directionless $1\times1\times1$ convolution disrupts this organization due to the absence of constraints on the information flow. Inspired by SMT (Lin et al., 2023), we introduce the Scale-Aware Integration Module (SAIM) in IsoNeXt block as a replacement for the inverted bottleneck. In the first expansion $1\times1\times1$ convolution layer, the convolution is divided into $s$ groups, consistent with the scale of the preceding group convolution. This structure ensures **intra-scale** feature fusion and learning. Subsequently, the features are reshaped, integrating one channel from each original group into new groups organized by scale. A grouped $1\times1\times1$ convolution with $C/s$ groups is then applied, where each group contains features from different scales, thus enforcing **inter-scale** feature fusion. This sequential intra-scale and inter-scale operation enables an effective multi-scale feature fusion and better learning of scale-aware information.

## 4. Experiments and Results

### 4.1. Experiment Settings

Datasets. This study uses two in-house datasets, both annotated for rectum and tumor regions, with all sensitive patient information de-identified. Dataset A, from Guangdong Provincial People's Hospital, China, contains 80 sagittal T2 MRI volumes of varying dimensions [18, 320, 320] to [24, 512, 512], with a median voxel spacing of [4.000, 0.875, 0.875]. It is randomly split into five folds for cross-validation, with results reported individually for each fold, as well as the mean and standard deviation across all folds. Dataset B, from the Second Affiliated Hospital of Navy Medical University, China, includes 77 sagittal T2 MRI volumes of fixed dimensions [21, 1024, 1024] and voxel spacing of [3.300, 0.234, 0.234]. It is used to evaluate models trained on Dataset A, assessing generalization to unseen data. During testing, images from Dataset B are resized to [21, 512, 512]. We further use a public dataset Prostate158 (Adams et al., 2022) for evaluation, which contains 158 biparametric 3T prostate MRI scans with expert annotations for two anatomical zones and tumor, officially divided into 119/20/19 for training/validation/testing.

Metrics. The Dice Coefficient serves as the primary evaluation metric. Statistical significance is assessed using a two-tailed t-test, with the p-value reported. For comparisons with state-of-the-art methods, the p-value is derived by comparing the proposed method's

performance with that of existing approaches. In ablation studies, the p-value is calculated by comparing baseline results to those obtained from modified configurations. We also use Hausdorff distance and Average surface distance as metrics for Prostate158.

Implementation details. The model is implemented in PyTorch and built on the framework of nnUNet, trained on a single NVIDIA RTX 3090 GPU. Images in Dataset A are resampled to a uniform voxel spacing of [4.000, 0.875, 0.875], then cropped into patches of [16, 320, 320] for 3D networks and [320, 320] for the 2D nnUNet. Training utilizes the SGD optimizer with a momentum of 0.99 for 100 epochs, starting with a learning rate of 0.01 and adjusted dynamically using a 'poly' decay strategy. The batch size is set to 2 for 3D networks and to 8 for the 2D nnUNet, while the scale factor for the Res2Net, Res2NeXt, and IsoNeXt blocks is set to 4. The model employs nnUNet's default data augmentation strategies and adopts the framework's standard composite loss function combining Cross Entropy and Dice losses.

## 4.2. Comparison with state-of-the-art Architectures.

In this section, we evaluate the performance of our proposed method against several state-of-the-art approaches across two datasets, as summarized in Table 1 and Table 2. The comparison includes convolution-based methods such as nnU-Net, 3D U-Net(Çiçek et al., 2016), Res-UNet(Zhang et al., 2018), and MedNeXt(Roy et al., 2023), as well as transformer-based methods like UNETR, SwinUNETR and nnFormer(Zhou et al., 2023). Additionally, we compare with a state-space model SegMamba, and MNet(Dong et al., 2022), specifically designed for anisotropic medical image segmentation. Except for 3D U-Net, all methods are implemented using their official code. Res-UNet is implemented via the MONAI framework(Cardoso et al., 2022).

Table 1: Evaluation on Dataset A (Dice (%)). The **best** and second best are highlighted.

| Model | #Para(M) | Fold0 | | Fold1 | | Fold2 | | Fold3 | | Fold4 | | Avg | |
|---|---|---|---|---|---|---|---|---|---|---|---|---|---|
| | | Rect | Tumor | Rect | Tumor | Rect | Tumor | Rect | Tumor | Rect | Tumor | Rectum | Tumor |
| 2D nnUNet | 47.63 | 68.65 | 49.81 | 67.76 | 52.10 | 72.06 | 45.20 | 67.66 | 45.70 | **70.55** | 54.08 | $69.34 \pm 1.91_{(p<0.05)}$ | $49.38 \pm 3.90_{(p<0.01)}$ |
| 3D nnUNet | 64.01 | 71.13 | 57.55 | 68.62 | 60.55 | 73.70 | 53.98 | 69.45 | 59.78 | 69.96 | 63.91 | $70.57 \pm 1.97_{(p=0.05)}$ | $59.15 \pm 3.38_{(p=0.07)}$ |
| 3D-UNet | 25.89 | 69.09 | 45.52 | 60.66 | 59.04 | 69.31 | 55.62 | 66.10 | 47.87 | 65.53 | 58.45 | $66.14 \pm 3.51_{(p<0.01)}$ | $53.30 \pm 6.22_{(p=0.01)}$ |
| Res-UNet | 19.22 | 67.11 | 55.50 | 62.46 | 50.96 | 70.59 | 48.15 | 67.50 | 50.21 | 67.94 | 58.66 | $67.12 \pm 2.94_{(p=0.01)}$ | $52.70 \pm 4.28_{(p<0.01)}$ |
| MNet | 8.78 | 69.44 | 57.62 | 61.15 | 50.00 | 68.64 | 48.78 | 65.97 | 52.44 | 68.02 | 59.08 | $66.64 \pm 3.33_{(p=0.01)}$ | $53.58 \pm 4.58_{(p<0.01)}$ |
| MedNeXt-B | 10.53 | 71.11 | 56.76 | 65.76 | **62.94** | 72.59 | 52.75 | 68.69 | 58.76 | 68.95 | 64.33 | $69.42 \pm 2.60_{(p<0.05)}$ | $59.11 \pm 4.69_{(p=0.12)}$ |
| UNETR | 137.21 | 64.88 | 42.39 | 57.95 | 46.52 | 63.77 | 35.52 | 64.90 | 44.92 | 60.83 | 42.69 | $62.47 \pm 3.02_{(p<0.01)}$ | $42.41 \pm 4.21_{(p<0.01)}$ |
| SwinUNETR | 38.30 | 68.43 | 55.59 | 61.95 | 57.91 | 69.36 | 53.91 | 66.34 | 50.58 | 61.80 | 52.41 | $65.58 \pm 3.55_{(p<0.01)}$ | $54.08 \pm 2.83_{(p<0.01)}$ |
| nnFormer | 149.17 | 69.00 | 52.42 | 65.78 | 57.42 | 73.63 | 54.28 | 67.54 | 49.41 | 67.67 | 61.07 | $68.73 \pm 2.97_{(p<0.01)}$ | $54.92 \pm 4.51_{(p<0.01)}$ |
| SegMamba | 67.36 | 68.93 | 54.48 | 61.81 | 52.46 | 71.08 | 48.20 | 65.91 | 48.92 | 64.84 | 52.77 | $66.51 \pm 3.60_{(p<0.01)}$ | $51.37 \pm 2.69_{(p<0.01)}$ |
| Ours | **2.76** | **72.64** | **62.35** | **69.70** | 61.52 | **76.01** | **64.49** | **69.88** | **61.23** | 70.03 | **67.51** | $\mathbf{71.65 \pm 2.72}$ | $\mathbf{63.42 \pm 2.62}$ |

The experimental results demonstrate that our method achieves state-of-the-art performance with a significantly lower model complexity. On Dataset A, our model outperforms all baselines, achieving the highest average Dice scores for rectum (71.65 ± 2.72) and tumor (63.42 ± 2.62), surpassing nnUNet 3D by 1.74% and 4.27%, respectively, with only 2.76 million parameters. This efficiency, along with robust performance across folds, highlights its effectiveness. On Dataset B, the model trained on Dataset A without retraining achieves competitive results, with Dice scores of 68.29 ± 1.24 for rectum and 45.45 ± 4.75 for tumor, outperforming 3D nnUNet by 0.50% and 4.92%, respectively. These results emphasize the model's strong generalization and computational efficiency. As for Prostate158, our method

achieved the highest Dice scores in both anatomical (including the Central Gland and Peripheral Zone) and tumor regions, while also demonstrating notable advantages in terms of Hausdorff distance and average surface distance.

Table 2: Comparison with state-of-the-art methods on Dataset B.

| Model | #Para(M) | Fold0 | | Fold1 | | Fold2 | | Fold3 | | Fold4 | | Avg | |
| --- | --- | --- | --- | --- | --- | --- | --- | --- | --- | --- | --- | --- | --- |
| | | Rect | Tumor | Rect | Tumor | Rect | Tumor | Rect | Tumor | Rect | Tumor | Rectum | Tumor |
| 2D nnUNet | 47.63 | 62.07 | 35.28 | 61.93 | 31.26 | 63.48 | 39.02 | 61.48 | 30.07 | 60.64 | 35.14 | $61.92 \pm 1.04_{(p<0.01)}$ | $34.15 \pm 3.57_{(p<0.01)}$ |
| 3D nnUNet | 64.01 | 68.71 | 42.00 | 66.58 | 36.62 | 68.56 | 41.90 | 68.26 | 37.26 | **66.86** | 44.87 | $67.79 \pm 1.00_{(p=0.10)}$ | $40.53 \pm 3.50_{(p<0.01)}$ |
| 3D-UNet | 25.89 | 61.13 | 15.89 | 55.21 | 13.59 | 60.17 | 20.52 | 63.05 | 19.90 | 55.76 | 29.35 | $59.06 \pm 3.43_{(p<0.01)}$ | $19.85 \pm 6.03_{(p<0.01)}$ |
| Res-UNet | 19.22 | 62.08 | 38.43 | 59.11 | 31.93 | 57.69 | 37.77 | 62.71 | 31.61 | 57.39 | 39.30 | $59.80 \pm 2.47_{(p<0.01)}$ | $35.81 \pm 3.73_{(p<0.01)}$ |
| MNet | 8.78 | 58.61 | 36.97 | 59.27 | 37.62 | 59.16 | 39.80 | 60.09 | 34.98 | 59.63 | 37.96 | $59.35 \pm 0.55_{(p<0.01)}$ | $37.47 \pm 1.74_{(p<0.01)}$ |
| MedNeXt-B | 10.53 | 66.46 | 40.94 | 64.12 | 34.76 | 65.92 | 44.00 | 66.15 | 37.43 | 64.63 | 43.49 | $65.46 \pm 1.02_{(p<0.01)}$ | $40.12 \pm 3.97_{(p<0.01)}$ |
| UNETR | 137.21 | 58.57 | 20.31 | 56.85 | 16.88 | 59.05 | 21.66 | 58.97 | 21.28 | 56.26 | 20.66 | $57.94 \pm 1.29_{(p<0.01)}$ | $20.16 \pm 1.91_{(p<0.01)}$ |
| SwinUNETR | 38.30 | 61.90 | 37.45 | 58.54 | 27.62 | 60.50 | 33.38 | 59.68 | 32.24 | 57.48 | 26.67 | $59.62 \pm 1.71_{(p<0.01)}$ | $31.48 \pm 4.41_{(p<0.01)}$ |
| nnFormer | 149.17 | 63.59 | 28.53 | 63.31 | 20.09 | 64.79 | 29.84 | 64.11 | 22.65 | 63.74 | 25.11 | $63.91 \pm 0.57_{(p<0.01)}$ | $25.24 \pm 4.04_{(p<0.01)}$ |
| SegMamba | 67.36 | 61.23 | 37.49 | 56.90 | 29.68 | 57.95 | 37.44 | 61.65 | 32.30 | 51.79 | 34.14 | $57.90 \pm 3.98_{(p<0.01)}$ | $34.21 \pm 3.37_{(p<0.01)}$ |
| Ours | **2.76** | **69.59** | **48.08** | **67.57** | **43.04** | **69.06** | **48.57** | **68.72** | **38.19** | 66.51 | **49.36** | **$68.29 \pm 1.24$** | **$45.45 \pm 4.75$** |

Table 3: Evaluation on Prostate158,HD-Hausdorff distance,ASD-average surface distance

| Model | #Para(M) | Central gland | | | Peripheral zone | | | Prostate tumor | | |
| --- | --- | --- | --- | --- | --- | --- | --- | --- | --- | --- |
| | | Dice | HD | ASD | Dice | HD | ASD | Dice | HD | ASD |
| 2D nnUNet | 47.63 | 86.88 | 13.37 | **0.98** | 75.59 | 15.01 | 0.99 | 35.88 | 26.42 | 6.55 |
| 3D nnUNet | 64.01 | 88.58 | **10.41** | 1.06 | 78.00 | 14.69 | 1.06 | 50.92 | 21.79 | 7.38 |
| 3D-UNet | 25.89 | 84.65 | 15.29 | 1.45 | 73.58 | 17.50 | 1.26 | 46.36 | 42.11 | 5.98 |
| Res-UNet | 19.22 | 85.90 | 11.81 | 1.28 | 73.23 | 17.51 | 1.31 | 47.47 | 32.58 | 4.29 |
| MNet | 8.78 | 87.73 | 11.52 | 0.99 | 75.56 | 13.99 | 1.01 | 52.58 | 29.88 | 7.61 |
| MedNeXt-B | 10.53 | 87.52 | 11.14 | 1.19 | 75.40 | 15.06 | 1.20 | 50.07 | 39.08 | 5.71 |
| UNETR | 137.21 | 86.18 | 17.77 | 1.32 | 70.07 | 18.77 | 1.60 | 40.25 | 52.00 | 4.92 |
| SwinUNETR | 38.30 | 84.82 | 18.81 | 1.28 | 70.67 | 18.17 | 1.39 | 48.57 | 37.78 | 3.69 |
| nnFormer | 149.17 | 86.58 | 13.85 | 1.18 | 74.68 | 17.46 | 1.25 | 48.06 | 44.61 | **3.38** |
| SegMamba | 67.36 | 86.84 | 11.96 | 1.16 | 73.68 | 19.05 | 1.26 | 49.80 | 40.30 | 4.34 |
| Ours | **2.76** | **88.96** | 10.50 | 1.02 | **79.43** | **13.02** | **0.90** | **53.00** | **20.70** | 3.97 |

## 4.3. Ablation Study

We compare the proposed architecture with several alternatives on Dataset A, as summarized in Table 4. The baseline model (first row) modifies the 3D-UNet architecture by integrating anisotropic kernels in the anisotropic stages and replacing BN with IN. The comparative analysis in the first two rows validates the effectiveness of the proposed AniNeXt block. In subsequent experiments, the anisotropic stages are fixed to employ the AniNeXt block, while the impact of various blocks in the isotropic stages is systematically evaluated. As shown in Table 4, all tested blocks improve performance on tumor segmentation, with transformer-inspired designs delivering particularly notable gains. In contrast, Res2Net demonstrates limited efficiency and performance, including a notable drop in accuracy for rectum segmentation, highlighting its limitations compared to the transformer-inspired pipeline. The best performance is achieved using AniNeXt with the IsoNeXt backbone, which outperforms other configurations, achieving segmentation accuracies of 71.65% for tumors and 67.51% for rectum, both statistically significant improvements. The comparison

between IsoNeXt and Res2NeXt in the last two rows further underscores the effectiveness of the proposed SAIM module, as the inclusion of SAIM is the distinction between the two.

Table 4: Ablation study on the use of different blocks.

| Aniso stage | Iso stage | #P(M) | Fold0 | | Fold1 | | Fold2 | | Fold3 | | Fold4 | | Avg | |
|---|---|---|---|---|---|---|---|---|---|---|---|---|---|---|
| | | | Rect | Tumor | Rect | Tumor | Rect | Tumor | Rect | Tumor | Rect | Tumor | Rect | Tumor |
| Conv | Conv | 22.14 | 69.68 | 60.67 | 64.12 | 58.36 | 73.11 | 52.77 | 68.67 | 55.97 | 66.74 | 62.33 | $68.46 \pm 3.35$ | $58.02 \pm 3.79$ |
| AniNeXt | Conv | 22.07 | 71.51 | 61.55 | 66.57 | 58.96 | 74.80 | 56.98 | 67.88 | 57.77 | 69.53 | 63.83 | $70.06 \pm 3.232_{(p=0.06)}$ | $59.82 \pm 2.83_{(p<0.05)}$ |
| AniNeXt | ConvNeXt | 5.54 | **72.64** | 61.40 | 66.85 | **61.60** | 75.14 | 61.23 | 69.65 | 58.68 | 68.52 | 64.75 | $70.56 \pm 3.32_{(p<0.01)}$ | $61.53 \pm 2.16_{(p=0.05)}$ |
| AniNeXt | Res2Net | 8.11 | 70.24 | 58.95 | 64.18 | 59.50 | 72.83 | 56.41 | 68.46 | 60.88 | 67.06 | 65.50 | $68.55 \pm 3.26_{(p=0.60)}$ | $60.25 \pm 3.35_{(p=0.13)}$ |
| AniNeXt | Res2NeXt | 5.52 | 71.08 | **62.67** | 68.33 | 60.04 | 75.35 | 61.87 | 69.77 | **61.33** | **70.25** | 64.31 | $70.96 \pm 2.65_{(p=0.01)}$ | $62.04 \pm 1.59_{(p<0.05)}$ |
| AniNeXt | IsoNeXt | 2.76 | **72.64** | 62.35 | **69.70** | 61.52 | **76.01** | **64.49** | **69.88** | 61.23 | 70.03 | **67.51** | $\mathbf{71.65 \pm 2.72}_{(p=0.01)}$ | $\mathbf{63.42 \pm 2.62}_{(p<0.05)}$ |

### 4.4. Visualization

We present qualitative comparisons between our method and other techniques on Dataset A, as shown in Fig. 4. In the first row, our method accurately predicts both the rectum and tumor, while other methods exhibit false positives in the tumor or incomplete rectum segmentation. In the second row, our method precisely predicts the tumor without displaying false positive discontinuities, unlike other methods. Overall, the visualizations demonstrate the superior tumor prediction performance of our method, with the performance on adjacent areas highlighting the effectiveness of our multi-scale design.

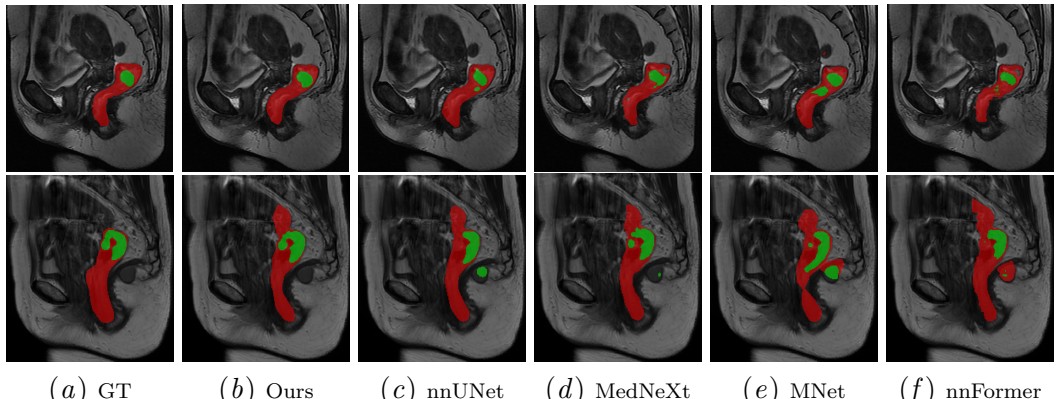

(a) GT    (b) Ours    (c) nnUNet    (d) MedNeXt    (e) MNet    (f) nnFormer

Figure 4: Qualitative comparison. Red region denotes rectum and green denotes tumor.

### 5. Conclusion and Discussion

This study introduces RCSegNeXt, a novel framework for rectal cancer segmentation in sagittal MRI scans. By integrating anisotropic and isotropic processing through the AniNeXt and IsoNeXt blocks with a Scale-Aware Integration Module (SAIM), the method enhances multi-scale feature representation effectively. Experimental results on two in-house datasets demonstrate significant improvements in tumor and rectum segmentation. Evaluation on public prostate dataset further verify the effectiveness. Future work will aim to enhance generalization across diverse datasets and explore applications in downstream tasks.

## Acknowledgments

This research is supported by Natural Science Foundation of China under Grant 62271465, Suzhou Basic Research Program under Grant SYG202338, and Open Fund Project of Guangdong Academy of Medical Sciences, China (No. YKY-KF202206).

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

## Appendix A. Selection of Kernels and Stride

The termination criterion for using anisotropic kernels is determined by the original spacing of the MRI scans. We limit the use of max-pooling operations to four, resulting in five stages (including the saddle layer). Given the original spacings $[s_z, s_x, s_y]$, the kernel sizes for convolutional stages and the strides for max-pooling operations in the encoder layers, as well as the saddle layer, are computed using Algorithm 1. The kernel sizes and strides for the decoder are designed as the mirror counterparts of the encoder. For example, when processing the spacing [4.000,0.875,0.875] from Dataset A, the computed strides are [[1,2,2],[1,2,2],[2,2,2],[2,2,2]], and the corresponding kernel sizes are [[1,3,3],[1,3,3],[3,3,3],[3,3,3],[3,3,3]]. These configurations align with the architecture depicted in Fig. 2.

---

**Algorithm 1:** Stride and Kernel Computation for Spacing Ratios

---

**Input:** $spacings = [s_z, s_x, s_y]$ (initial spacing values)
**Output:** $strides$, $kernels$ (lists of computed strides and kernels)
$strides \leftarrow \emptyset$, $kernels \leftarrow \emptyset$;
**for** $i \leftarrow 1$ **to** $4$ **do**
    Compute $spacing\_ratio \leftarrow [sp/\min(spacings)$ for $sp \in spacings]$;
    Compute $stride \leftarrow [2$ if $ratio \leq 2$ else $1$ for $ratio \in spacing\_ratio]$;
    Compute $kernel \leftarrow [3$ if $ratio \leq 2$ else $1$ for $ratio \in spacing\_ratio]$;
    Update $spacings \leftarrow [sp \cdot st$ for $sp, st \in zip(spacings, stride)]$;
    Append $stride$ to $strides$;
    Append $kernel$ to $kernels$;
**end**
Compute $spacing\_ratio \leftarrow [sp/\min(spacings)$ for $sp \in spacings]$;
Compute $kernel \leftarrow [3$ if $ratio \leq 2$ else $1$ for $ratio \in spacing\_ratio]$;
Append $kernel$ to $kernels$;
**return** $strides$, $kernels$;

---

## Appendix B. Other Analysis

### B.1. Built of Baseline

The baseline used in the ablation study, as presented in Table 4, incorporates several modifications to the standard 3D-UNet architecture. In this section, we analyze the effects of these modifications in detail, with the results summarized in Table 5. From the table, it is evident that implementing the anisotropic-isotropic design consistently improves performance across both the rectum and tumor segmentation tasks. However, when Batch Normalization (BN) is replaced with Instance Normalization (IN), performance improves for tumor segmentation but decreases for rectum segmentation. This discrepancy can be attributed to the significant heterogeneity in tumor characteristics and inter-individual variations, as opposed to the relatively consistent morphology and anatomical positioning of the rectum across different individuals.The baseline in Table 4 outperforms several sota methods in Ta-

ble 1 mainly stems from the two design strategies described above while they don't adopt these.

Table 5:  Built of baseline for ablation study.

| Model | Fold0 | | Fold1 | | Fold2 | | Fold3 | | Fold4 | | Avg | |
|---|---|---|---|---|---|---|---|---|---|---|---|---|
| | Rect | Tumor | Rect | Tumor | Rect | Tumor | Rect | Tumor | Rect | Tumor | Rectum | Tumor |
| 3D-UNet | 69.09 | 45.52 | 60.66 | 59.04 | 69.31 | 55.62 | 66.10 | 47.87 | 65.53 | 58.45 | $66.14 \pm 3.51$ | $53.30 \pm 6.22$ |
| →Aniso-iso design | 69.98 | 55.43 | 64.31 | 59.31 | 75.21 | 54.06 | 68.24 | 50.31 | 68.24 | 56.95 | $69.20 \pm 3.95$ | $55.21 \pm 3.36$ |
| BN→IN | 69.68 | 60.67 | 64.12 | 58.36 | 73.11 | 52.77 | 68.67 | 55.97 | 66.74 | 62.33 | $68.46 \pm 3.35$ | $58.02 \pm 3.79$ |

## B.2. Comparison with Variants of state-of-the-art Methods

In this study, we compare several variants of state-of-the-art methods, as presented in Table 6 and Table 7. The variants include MedNeXt models with different sizes and kernel configurations, as well as UNETR models with varying patch sizes for tokenization. As shown in Table 6 and Table 7, increasing the convolutional kernel size does not lead to improved performance. This can be attributed to the fact that the identification of rectal cancer-related anatomy, particularly tumors, demands finer details. Therefore, merely enlarging the receptive field is insufficient for this task. Instead, multi-scale feature extraction is required, which further supports the effectiveness of our proposed method. For the transformer-based architecture UNETR, reducing the patch size along the Z-axis improves the recognition of relevant structures, particularly tumors. However, its performance remains inferior to other convolution-based methods. This is primarily due to the limited number of tokens along the Z-axis—4 for a patch size of 4×16×16 and only 1 for 16×16×16—which significantly restricts the model's ability to learn fine details.

Table 6: Comparison with variants of state-of-the-art methods on Dataset A.

| Model | #Para(M) | Fold0 | | Fold1 | | Fold2 | | Fold3 | | Fold4 | | Avg | |
|---|---|---|---|---|---|---|---|---|---|---|---|---|---|
| | | Rect | Tumor | Rect | Tumor | Rect | Tumor | Rect | Tumor | Rect | Tumor | Rectum | Tumor |
| MedNeXt-B(k=3) | 10.53 | 71.11 | 56.76 | 65.76 | 62.94 | 72.59 | 52.75 | 68.69 | 58.76 | 68.95 | 64.33 | $69.42 \pm 2.60_{(p<0.05)}$ | $59.11 \pm 4.69_{(p=0.12)}$ |
| MedNeXt-B(k=5) | 10.96 | 68.51 | 55.64 | 65.48 | 60.00 | 72.48 | 56.84 | 67.76 | 55.87 | 67.57 | 60.17 | $68.36 \pm 2.56_{(p<0.01)}$ | $57.70 \pm 2.22_{(p<0.01)}$ |
| MedNeXt-L(k=3) | 61.78 | 70.16 | 56.02 | 67.11 | 64.24 | 72.21 | 57.24 | 67.83 | 58.26 | 70.26 | 63.05 | $69.51 \pm 2.05_{(p<0.05)}$ | $59.76 \pm 3.66_{(p=0.11)}$ |
| MedNeXt-L(k=5) | 62.99 | 71.06 | 57.78 | 67.42 | 61.19 | 70.37 | 54.14 | 68.38 | 55.09 | 68.60 | 65.39 | $69.17 \pm 1.50_{(p<0.05)}$ | $58.72 \pm 4.63_{(p=0.05)}$ |
| UNETR(16×16×16) | 146.51 | 67.12 | 35.18 | 58.24 | 49.03 | 63.53 | 32.80 | 63.18 | 36.63 | 61.28 | 34.90 | $62.67 \pm 3.25_{(p<0.01)}$ | $37.71 \pm 6.48_{(p<0.01)}$ |
| UNETR(4×16×16) | 137.21 | 64.88 | 42.39 | 57.95 | 46.52 | 63.77 | 35.52 | 64.90 | 44.92 | 60.83 | 42.69 | $62.47 \pm 3.02_{(p<0.01)}$ | $42.41 \pm 4.21_{(p<0.01)}$ |
| Ours | 2.76 | 72.64 | 62.35 | 69.70 | 61.52 | 76.01 | 64.49 | 69.88 | 61.23 | 70.03 | 67.51 | $71.65 \pm 2.72$ | $63.42 \pm 2.62$ |

Table 7: Comparison with variants of state-of-the-art methods on Dataset B.

| Model | #Para(M) | Fold0 | | Fold1 | | Fold2 | | Fold3 | | Fold4 | | Avg | |
|---|---|---|---|---|---|---|---|---|---|---|---|---|---|
| | | Rect | Tumor | Rect | Tumor | Rect | Tumor | Rect | Tumor | Rect | Tumor | Rectum | Tumor |
| MedNeXt-B(k=3) | 10.53 | 66.46 | 40.94 | 64.12 | 34.76 | 65.92 | 44.00 | 66.15 | 37.43 | 64.63 | 43.49 | $65.46 \pm 1.02_{(p<0.01)}$ | $40.12 \pm 3.97_{(p<0.05)}$ |
| MedNeXt-B(k=5) | 10.96 | 65.30 | 39.16 | 61.54 | 32.25 | 63.73 | 44.98 | 65.32 | 37.31 | 61.32 | 39.46 | $63.44 \pm 1.95_{(p<0.01)}$ | $38.63 \pm 4.58_{(p<0.05)}$ |
| MedNeXt-L(k=3) | 61.78 | 68.20 | 42.68 | 65.40 | 38.37 | 69.24 | 49.75 | 66.77 | 43.25 | 67.72 | 49.16 | $67.47 \pm 1.46_{(p=0.28)}$ | $44.64 \pm 4.79_{(p=0.70)}$ |
| MedNeXt-L(k=5) | 62.99 | 67.18 | 40.26 | 63.55 | 33.98 | 65.25 | 43.19 | 66.76 | 35.56 | 62.29 | 37.33 | $65.01 \pm 2.08_{(p<0.01)}$ | $38.06 \pm 3.69_{(p<0.01)}$ |
| UNETR(16×16×16) | 146.51 | 56.66 | 16.43 | 52.54 | 16.14 | 57.71 | 19.53 | 56.76 | 19.35 | 52.04 | 17.18 | $55.14 \pm 2.64_{(p<0.01)}$ | $17.73 \pm 1.61_{(p<0.01)}$ |
| UNETR(4×16×16) | 137.21 | 58.57 | 20.31 | 56.85 | 16.88 | 59.05 | 21.66 | 58.97 | 21.28 | 56.26 | 20.66 | $57.94 \pm 1.29_{(p<0.01)}$ | $20.16 \pm 1.91_{(p<0.01)}$ |
| Ours | 2.76 | 69.59 | 48.08 | 67.57 | 43.04 | 69.06 | 48.57 | 68.72 | 38.19 | 66.51 | 49.36 | $68.29 \pm 1.24$ | $45.45 \pm 4.75$ |

## B.3. Effect of Model Scaling

This section investigates the impact of scaling the model and examines whether increasing model size leads to improved performance. To explore this hypothesis, we scale up the model and compare the results across different configurations. As depicted in Fig. 2, the model architecture comprises four downsampling operations, resulting in five stages. The resolution transitions from the finest, corresponding to the input scans, to the coarsest at the saddle layer, labeled as Stage 1 to Stage 5. The baseline model, denoted as RCSegNeXt-B (where "B" stands for the base model), consists of two repeated AniNeXt/IsoNeXt blocks per stage. To assess the impact of scaling, we increase the number of block repetitions from [2,2,2,2,2] (used in RCSegNeXt-B) to [2,2,2,8,4], yielding the RCSegNeXt-L model, where "L" represents the large-scale variant. This modification substantially increases the number of model parameters from 2.76M to 10.77M.

A comparative analysis of these models is presented in Table 8. On Dataset A, the Dice coefficient for the rectum exhibits a marginal improvement from 71.65 ± 2.72 to 71.78 ± 3.00, while for the tumor, it slightly decreases from 63.42 ± 2.62 to 62.46 ± 2.99. However, when directly evaluated on Dataset B, the Dice score consistently improves for both the rectum and the tumor. Specifically, the Dice score for the rectum increases from 68.29 ± 1.24 to 68.40 ± 1.22, and for the tumor, from 45.45 ± 4.75 to 47.39 ± 3.38. These trends align with the performance of the MedNeXt-B and MedNeXt-L models, as shown in Table 6 and Table 7. Notably, while the scaled-up models do not demonstrate consistent improvements in cross-validation on Dataset A, they achieve substantial performance gains in direct testing on Dataset B. We hypothesize that this discrepancy may stem from the relatively small sample size of Dataset A (N = 80). The smaller model may better capture the specific characteristics of this limited dataset, resulting in strong cross-validation performance. However, due to the domain gap between Dataset A and Dataset B, the smaller model may fail to generalize when tested on Dataset B, as its learned features might not be representative of the broader distribution. In contrast, the larger model, though potentially undertrained on the smaller dataset, possesses a greater capacity to learn complex and abstract features. This enables it to generalize more effectively, leading to superior performance on Dataset B, even if it does not significantly outperform the smaller model on Dataset A. These findings highlight the advantage of larger models in capturing more generalized feature representations, particularly when sufficient training data is available. Moving forward, we plan to collect additional retrospective data to further enhance the performance of the scaled-up architecture.

Table 8: Experiments on the impact of model scaling on Dataset A and B.

| Dattset | Model | #Para(M) | Fold0 | | Fold1 | | Fold2 | | Fold3 | | Fold4 | | Avg | |
|---|---|---|---|---|---|---|---|---|---|---|---|---|---|---|
| | | | Rect | Tumor | Rect | Tumor | Rect | Tumor | Rect | Tumor | Rect | Tumor | Rectum | Tumor |
| A | RCSegNeXt-B | 2.76 | **72.64** | **62.35** | 69.70 | **61.52** | 76.01 | 64.49 | 69.88 | **61.23** | 70.03 | **67.51** | 71.65 ± 2.72 | **63.42 ± 2.62** |
| | RCSegNeXt-L | 10.77 | 71.63 | 60.43 | **69.71** | 60.36 | **76.99** | **65.79** | **70.25** | 60.04 | **70.30** | 65.67 | **71.78 ± 3.00** | 62.46 ± 2.99 |
| B | RCSegNeXt-B | 2.76 | 69.59 | **48.08** | 67.57 | 43.04 | 69.06 | 48.57 | **68.72** | 38.19 | 66.51 | 49.36 | 68.29 ± 1.24 | 45.45 ± 4.75 |
| | RCSegNeXt-L | 10.77 | **69.88** | 47.82 | **68.01** | **44.94** | **69.26** | **49.41** | 68.15 | **43.17** | **66.72** | **51.60** | **68.40 ± 1.22** | **47.39 ± 3.38** |

### B.4. Analysis of Failure Cases

Herein, we present several representative failure cases of our our RCSegNeXt model in Fig. 5. The first row displays ground-truth, while the second row shows the corresponding predictions from our RCSegNeXt. Each column represents a distinct failure case. Cases 1-2 illustrate suboptimal segmentation performance at the superior rectal margin, highlighted by the yellow bounding boxes in Fig. 5(a) and Fig. 5(b), which can also be observed in the second row of Fig. 4. This issue arises from the indistinct anatomical boundary between the rectum and sigmoid colon, further exacerbated by inter-slice discontinuities that hinder precise segmentation of these contiguous structures. Cases 3–5 share a common characteristic: they correspond to tumor emergence or disappearance slices located at the sagittal boundary of the tumor. Our model exhibited performance degradation in these regions, a phenomenon also observed in the rectum. This phenomenon primarily also stems from relatively thick slice thickness, causing actual tumor/rectal boundaries to potentially lie between adjacent slices, thus making it difficult to definitively assign these boundaries to specific slices. To mitigate these issues, we intend to design two complementary strategies: (1) Two-Stage Architecture: The first stage segments the rectal region using merged rectum-tumor labels, while the second stage refines segmentation within a region of interest (ROI) cropped from the first-stage output. (2) Multi-Axial MRI Integration: Incorporating auxiliary inputs from orthogonal MRI planes provides high-resolution details between sagittal slices, offering "free lunch" information to enhance boundary definition.

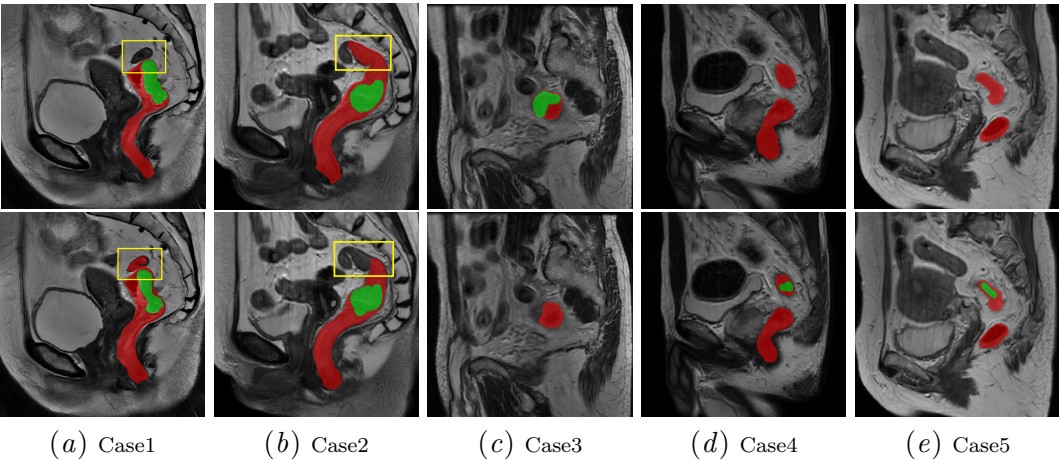

(a) Case1    (b) Case2    (c) Case3    (d) Case4    (e) Case5

Figure 5: Illustration of failure cases. Red region denotes rectum and green denotes tumor.

