# OpenReview forum: "RCSegNeXt: Efficient multi-scale ConvNeXt for rectal cancer segmentation from sagittal MRI scans"
_MIDL.io/2025/Conference — MIDL 2025 Poster_

### Official Review · Reviewer_krWy · 2025-02-09

**Confidence:** 4
**Preliminary Rating:** 4
**Recommendation:** Poster
**Final Rating:** 4

**Summary:**

This paper proposes an MRI segmentation network named RCSegNeXt, with its main contributions being two proposed network modules, AniNeXt and IsoNeXt. The proposed network outperforms common 2D and 3D medical segmentation methods on two internal datasets.

**Strengths:**

1. This paper addresses a relatively underexplored area, rectal cancer segmentation, contributing to the development of future related research.
2. The proposed network is lightweight, making it suitable for practical deployment while demonstrating satisfactory performance.

**Weaknesses:**

1. The technical novelty is limited. AniNeXt can be seen as a simple extension of ConvNeXt to three dimensions, while SAIM is similar to Scale-Aware Aggregation [*1].
2. The experiments are conducted only on in-house datasets, making it unclear whether the method can generalize to more public 3D medical datasets. It would be very helpful if these in-house datasets could be made public for academic use.
3. Some experimental details are unclear, such as the loss function.

[*1] Scale-Aware Modulation Meet Transformer, ICCV 23

**Detailed Comments:**

See weakness.

**Justification Of The Final Rating:**

After reading the comments of the other reviewers and the author's response, I think the problems with this paper have basically been properly addressed. Therefore, I will keep my score as ‘weak accept’.

**Justification Of The Preliminary Rating:**

Although the novelty of some proposed modules is relatively limited, the paper offers an interesting approach to achieving MRI rectal cancer segmentation. Therefore, my current recommendation is weak accept.

**Questions To Address In The Rebuttal:**

See weakness.

**Special Issue:**

No

---

> ### Author Response · Authors · 2025-03-08
>
> **In response to Reviewer #3(krWy):**
>
> We sincerely appreciate the insightful feedback provided by the Reviewer. Below, we address the specific concerns raised:
>
> **Comparison with Scale-Aware Aggregation:**
>
> We acknowledge that the IsoNeXt block draws inspiration from the ICCV 2023 work "Scale-Aware Modulation Meets Transformer." In response, we have cited this paper in **Section 3.3 (Page 6)**. However, our IsoNeXt block differs from the Scale-Aware Aggregation (SAA) module in several aspects. Specifically, our approach employs group convolution for intra-scale operations, ensuring that information interaction remains confined within each scale. In contrast, SAA does not utilize group convolution in this stage; instead, it applies a unified pointwise convolution to the aggregated features across all scales. By incorporating group convolution in the intra-scale stage, our method effectively mitigates interference between different scales, as scale interaction is explicitly handled in the subsequent step of the IsoNeXt block. Furthermore, group convolution enhances computational efficiency, making our approach more effective in multi-scale feature processing.
>
> **Generalization on Public Datasets:**
>
> To further assess the generalizability of our approach, we conducted additional validation on the publicly available Prostate158 dataset, as described in **Section 4.2 (Pages 7–8).** This dataset was selected primarily due to its imaging parameters and anatomical similarity to rectal cancer MRI, allowing for a meaningful assessment within the complex pelvic structures. Additionally, Prostate158 provides evaluation on both anatomical structures and challenging tumor regions, making it well-suited for comprehensive validation. We strictly adhered to the official data splits and experimental settings. Our model achieved the highest Dice scores for the Central Gland, Peripheral Zone, and Tumor regions, surpassing the second-best method by 0.38, 1.43, and 0.42 Dice points, respectively. Regarding Hausdorff Distance, our approach outperformed all others for the Peripheral Zone and Tumor regions and achieved the second-best performance for the Central Gland. For Average Surface Distance, our model ranked first for the Peripheral Zone and third for both the Central Gland and Tumor regions. These results further confirm the effectiveness of our proposed architecture.
> Additionally, we have **open-sourced** our code on [GitHub](https://github.com/WayneBo98/RCSegNeXt) to enhance reproducibility and facilitate further research. As for making our dataset publicly available for academic use, this process requires navigating complex ethical reviews and institutional approvals. We are actively working to secure permission from the two collaborating hospitals, and once obtained, we will release the dataset as soon as possible. In the meantime, researchers can utilize our publicly available model and checkpoint for direct inference or, more advancedly, perform source-free domain adaptation on their own rectal cancer datasets.
>
> **Clarification of Experimental Details:**
>
> To improve clarity, we have provided a more detailed description of our experimental setup in **Section 4.1 (Page 7).** Specifically, we utilized the nnUNet framework and adopted its default data augmentation and loss function configurations. The loss function consists of a combination of Cross-Entropy loss and Dice loss.
>
> Once again, we sincerely appreciate the Reviewer's valuable comments. We also greatly appreciate the reviewers' interest in the application of deep learning techniques in the field of rectal cancer diagnosis.

---

### Official Review · Reviewer_nzWC · 2025-02-09

**Confidence:** 3
**Preliminary Rating:** 3
**Recommendation:** Poster

**Summary:**

This paper proposes a novel U-shaped segmentation network for MRI segmentation, with two core contributions: the AniNeXt block and the IsoNeXt block. Experiments on two datasets demonstrate the superior performance of the proposed algorithm. This proposed RCSegNeXt is also the first MRI rectal cancer segmentation method.

**Strengths:**

1. MRI cancer segmentation is an important task, aligning well with the scope of MIDL.
2. The figures in this paper are well-designed and easy to understand.
3. The proposed method is simple and easy to implement.
4. Experiments on two datasets demonstrate the superiority of the proposed method.

**Weaknesses:**

1. The paper lacks a comprehensive review of related works in medical image analysis. It is recommended to add a separate "Related Work" section, categorically discussing common medical image segmentation algorithms, such as Transformer-based methods, Mamba-based methods, and foundational model(SAM1, SAM2)-based methods .
2. As shown in Table 3, even for the simplest baseline (using simple Conv for both Aniso and Iso stages), the network achieves a performance of 68.46, surpassing most comparison methods. This phenomenon requires further analysis and explanation.
3. The proposed method is only validated on internal datasets, making it unclear whether it is also effective on other public datasets.
4. It is recommended to present some failure cases and analyze potential directions for future improvements.

**Detailed Comments:**

See weakness.

**Justification Of The Preliminary Rating:**

Overall, the paper is well-written, but the related work and ablation experiments require further analysis and improvement. If the authors can address these concerns, I would consider adjusting my score.

**Questions To Address In The Rebuttal:**

See weakness.

**Special Issue:**

No

---

> ### Author Response · Authors · 2025-03-08
>
> **In response to Reviewer #2(nzWC):**
>
> We sincerely appreciate the reviewer’s constructive feedback and insightful comments. Below, we address the specific concerns raised:
>
> **Lack of a comprehensive review of related works:**
>
> To enhance the contextualization of our study, we have incorporated a new "Related Work" section (**Section 2, Page 3**) in the main text. This section provides a chronological overview of key architectural advancements in deep learning for medical image segmentation, covering CNN-based architectures, transformer-based models, state-space models, and foundation models. We have systematically introduced several milestone works, including UNet, nnUNet, TransUNet, SwinUNETR, SegMamba, and SAM, ensuring a structured and comprehensive review.
>
> **Concern regarding the baseline performance:**
>
> In **Appendix Section B.1 (Page 13)**, we have detailed the construction process of the baseline model of Table 4. As shown in **Table 5 (Page 14)**, the introduction of an anisotropic-isotropic design and the replacement of Batch Normalization with Instance Normalization led to performance improvements over the 3D UNet baseline, with Dice score gains of 2.32 and 4.72 for rectum and tumor segmentation, respectively. We emphasize that these design choices, particularly the anisotropic-isotropic strategy, help mitigate the thick-slice problem, which is often overlooked by state-of-the-art (SOTA) methods, as they are often initially designed for isotropic scans, typically with spacings of 1.0×1.0×1.0 (mm).
>
> **Validation beyond internal datasets:**
>
> To further evaluate the generalizability of our approach, we validated it on the public Prostate158 dataset, as detailed in **Section 4.2 (Pages 7-8)**. This dataset was selected primarily due to its imaging parameters and anatomical similarity to rectal cancer MRI, allowing for a meaningful assessment within the complex pelvic structures. Additionally, Prostate158 provides evaluation on both anatomical structures and challenging tumor regions, making it well-suited for comprehensive validation. We strictly followed the official data splits and experimental settings. Our model achieved the best Dice scores for the Central Gland, Peripheral Zone, and Tumor regions, surpassing the second-best method by 0.38, 1.43, and 0.42 Dice points, respectively. In terms of Hausdorff Distance, we achieved the best performance for the Peripheral Zone and Tumor, and the second-best for the Central Gland. For Average Surface Distance, our model ranked first for the Peripheral Zone and third for the Central Gland and Tumor. These results further demonstrate the effectiveness of our proposed architecture. Additionally, we have **open-sourced** our code on [GitHub](https://github.com/WayneBo98/RCSegNeXt) to promote reproducibility and facilitate further research.
>
> **Showcasing failure cases:**
>
> In **Appendix B.4 (Page 16)**, we present five representative failure cases, which primarily fall into two categories:
>
> - Inferior performance at the superior margin of the rectum
>
> - Suboptimal segmentation near the emergence and disappearance slices along the sagittal boundary
>
> Both issues are likely influenced by slice thickness, with the first also attributed to the ambiguous boundary between adjacent anatomies. To address these challenges, we plan to incorporate the following strategies in future work:
>
> - A two-stage network: First, a localization network will identify the rectum region, followed by a refinement stage to improve segmentation accuracy for both the rectum and tumor. This coarse-to-fine approach can enhance overall performance.
>
> - Multi-axial MRI integration: Leveraging orthogonal MRI planes can provide high-resolution details between sagittal slices, offering a form of “free lunch” to improve segmentation quality.
>
> Once again, we appreciate the reviewer’s valuable feedback, which has significantly contributed to strengthening our work.

---

### Official Review · Reviewer_Ng6E · 2025-02-21

**Confidence:** 5
**Preliminary Rating:** 4
**Recommendation:** Poster
**Final Rating:** 5

**Summary:**

The authors present RCSegNeXt, a multiscale ConvNeXt-based deep learning architecture for rectal cancer segmentation in sagittal MRI scans. They introduce 2 block designs - AniNeXt and IsoNeXt blocks - the former specifically designed for anisotropic image and the later for mutliscale feature learning. The authors evaluate on 2 in-house datasets with extensive comparison against state-of-the-art baselines.

**Strengths:**

1. The authors present a well written paper with clear motivations and descriptions of their methods. Their proposed modules are backed up by ablation experiments with consistent benefits of using the modules.
2. The authors use a number of baselines to compare against their network and demonstrate superior performance.

**Weaknesses:**

1. The authors’ choices of baselines do make their evaluation slightly lacking. For example, they use the architecture UNETR while foregoing the comparatively superior architecture SwinUNETR. They could also compare against better architectures such as CoTr or MedNeXt-L but just the SwinUNETR comparison could be beneficial.
2. The authors could benefit from scaling experiments for their architecture, but this is merely a suggestion.

**Detailed Comments:**

Please check the Strengths, Weaknesses and Rebuttal sections.

**Justification Of The Final Rating:**

The authors already had decent empirical results which they augmented with multiple experiments in the rebuttal. I consider this effective in demonstrating the benefits of their proposed architecture.

**Justification Of The Preliminary Rating:**

The authors propose a decent architecture which demonstrable benefits for their proposed module. The experiments are sufficient against a number of well-known baselines and the paper is very clearly written.

**Questions To Address In The Rebuttal:**

The rebuttal could address the inclusion of newer and slightly better baselines as mentioned in the weaknesses.

---

> ### Author Response · Authors · 2025-03-08
>
> **In response to Reviewer #1(Ng6E):**
>
> We sincerely appreciate the Reviewer's positive feedback and insightful comments. Below, we address the primary concern raised in the review.
>
> **Inclusion of Newer and More Competitive Baselines:**
>
> In response to the Reviewer's valuable suggestion, we have incorporated SwinUNETR as an additional baseline in our comparative analysis, as presented in **Table 1 and Table 2 (Pages 7 and 8)** of the main text. Our experimental results demonstrate that SwinUNETR outperforms UNETR, achieving Dice score improvements of 3.11 and 11.67 on Dataset A (Table 1) and 1.68 and 11.32 on Dataset B (Table 2). Nevertheless, our proposed method continues to exhibit superior performance, surpassing SwinUNETR by 6.07 and 9.34 Dice points on Dataset A and by 8.67 and 13.97 Dice points on Dataset B. Furthermore, we have extended our analysis in **Appendix Section B.2 (Page 14)** by including comparisons with multiple state-of-the-art (SOTA) variants, specifically MedNeXt-B with kernel sizes of 3 and 5, as well as MedNeXt-L with kernel sizes of 3 and 5. Additionally, we examine UNETR using different patch sizes, including 4×16×16 (as reported in the main text) and 16×16×16. Our findings indicate that while MedNeXt-L (k=3) slightly outperforms MedNeXt-B (k=3) on Dataset A, it demonstrates stronger generalization capability on Dataset B. Notably, increasing the kernel size does not necessarily lead to performance improvements, as the features critical to this task are not strictly dependent on a large receptive field but rather on multi-scale feature extraction. Moreover, our comparison of different patch sizes in UNETR highlights the importance of capturing fine-grained details. This observation also elucidates the substantial performance gap between UNETR and SwinUNETR, as SwinUNETR is more effective at preserving fine-grained structural information than UNETR.
>
> **Scaling of the architecture**
>
> Following the Reviewer's recommendation, we performed several scaling experiments, as shown in **Appendix Section B.3 (Page 15)**. Specifically, we scaled up our model by implementing 8× IsoNeXt blocks in the fourth stage and 4× IsoNeXt blocks in the fifth stage. Consequently, the block repetition configuration was adjusted from [2,2,2,2,2] to [2,2,2,8,4], leading to an increase in model parameters from 2.76M to 10.77M. We refer to the smaller model presented in the main text as RCSegNeXt-B and the scaled-up variant as RCSegNeXt-L. However, the performance improvements were not consistent across all metrics. On Dataset A, the Dice coefficient for rectum segmentation showed a marginal increase from 71.65±2.72 to 71.78±3.00 with RCSegNeXt-L, while the Dice coefficient for tumor segmentation slightly decreased from 63.42±2.62 to 62.46±2.99. In contrast, Dataset B exhibited a consistent performance improvement, with the Dice score for rectum increasing from 68.29±1.24 to 68.40±1.22 and for tumor rising from 45.45±4.75 to 47.39±3.38. A similar trend was observed when scaling MedNeXt-B to MedNeXt-L, where only marginal improvements were achieved on Dataset A for both rectum and tumor segmentation, whereas significant gains were observed on Dataset B. This discrepancy may stem from the relatively small size of Dataset A, where smaller models tend to fit better to dataset-specific characteristics that do not generalize well to Dataset B due to domain shifts. In contrast, larger models can learn more complex and abstract feature representations, thereby exhibiting stronger generalization capabilities even though not sufficiently trained on the small source dataset. To further enhance the performance of the scaled-up architecture, we plan to collect additional retrospective data in future studies.
>
> We again appreciate the Reviewer's suggestion, which has helped us strengthen our analysis and provide a more comprehensive evaluation.

---

> > ### Comment · Reviewer_Ng6E · 2025-03-13
> > **Reply to rebuttal.**
> >
> > I thank the authors in their effort for the rebuttal. I will increase my rating in response.

---

> > > ### Author Response · Authors · 2025-03-14
> > >
> > > We sincerely appreciate your positive feedback and the time you have taken to review our work. Thank you for your constructive comments and support!

---

### Author Rebuttal · Authors · 2025-03-07

**Rebuttal:**

We would like to thank all the Reviewers and the Chairs for their hard work. We have carefully read the reviews and addressed all the issues raised, which undoubtedly improved the quality of our submission.

We uploaded the revised version of our work and are ready to participate in further discussion. Based on comments, we have made the following changes to the revised version, as highlighted in our newest manuscript submitted in the rebuttal stage. A more detailed version will be elaborated upon in the comments provided to each reviewer:

**In Response to Common Concerns of Reviewers**
- To address Reviewer #2 and Reviewer #3’s concern about using only private datasets, we further validated our approach on a public prostate dataset Prostate158, as detailed in **Section 4.2**  on **Page 8** of the main text. We further open-sourced our code on [GitHub](https://github.com/WayneBo98/RCSegNeXt).

**In Response to Reviewer #1 (Ng6E):**
- At Reviewer #1’s suggestion, **Tables 1, 2, and 3** on **Page 7 and Page 8** of the main text now include SwinUNETR for comparison. Additionally, **Appendix Section B.2 (Page 14)** now includes comparisons against multiple SOTA variants, such as MedNeXt-L.
- In response to Reviewer #1’s valuable suggestion, we performed a scaling-up experiment and provided a detailed analysis, as shown in **Appendix Section B.3 (Page 15)**.

**In Response to Reviewer #2 (nzWC):**
- To address Reviewer #2’s concern on the lack of related work, we have added a new "Related Work" section (**Section 2**) on **Page 3** of the main text.
- To address Reviewer #2’s concern on the performance of the baseline, in **Page 13 of Appendix Section B.1**, we have provided the detailed construction process of the baseline from Table 4, along with an explanation.
- In response to Reviewer #2’s valuable comments, we have provided representative failure cases on **Page 16 of Appendix B.4**, along with an analysis of potential causes and possible solutions.

**In Response to Reviewer #3 (krWy):**
- In response to Reviewer #3’s valuable comments, we have added the ICCV2023 paper into citation in **Section 3.3 (Page 6)**.
- In response to the suggestions provided by Reviewer #3, we have added a detailed description of the experimental details of the framework and loss function used in **Section 4.1 (Page 7)**.

We thank the Reviewers once again and look forward to discussing any other aspects of the paper that require further clarification.

---
**Authors**

**Supporting Material:**

/attachment/47eaea46a9f028cf85a1a7a7db75a94c1b3e5fd8.pdf

---

### Meta-Review · Area_Chair_65NJ · 2025-03-18

**Recommendation:** Accept (Poster)
**Confidence:** 5

**Metareview:**

The authors have addressed the concerns raised by the reviewers with additional experiments in the rebuttal. One reviewer mentioned the limited novelty of the work and maintained the rating after the rebuttal. I recommend acceptance as a poster.